# Molecular mechanism of active Cas7-11 in processing CRISPR RNA and interfering target RNA

**Hemant N Goswami[1], Jay Rai[1], Anuska Das[1], Hong Li[1,2]***

[1]Institute of Molecular Biophysics, Florida State University, Tallahassee, United States; [2]Department of Chemistry and Biochemistry, Florida State University, Tallahassee, United States

**Abstract** Cas7-11 is a Type III-E CRISPR Cas effector that confers programmable RNA cleavage and has potential applications in RNA interference. Cas7-11 encodes a single polypeptide containing four Cas7- and one Cas11-like segments that obscures the distinction between the multi-subunit Class 1 and the single-subunit Class-2 CRISPR Cas systems. We report a cryo-EM (cryo-electron microscopy) structure of the active Cas7-11 from *Desulfonema ishimotonii* (DiCas7-11) that reveals the molecular basis for RNA processing and interference activities. DiCas7-11 arranges its Cas7- and Cas11-like domains in an extended form that resembles the backbone made up by four Cas7 and one Cas11 subunits in the multi-subunit enzymes. Unlike the multi-subunit enzymes, however, the backbone of DiCas7-11 contains evolutionarily different Cas7 and Cas11 domains, giving rise to their unique functionality. The first Cas7-like domain nearly engulfs the last 15 direct repeat nucleotides in processing and recognition of the CRISPR RNA, and its free-standing fragment retains most of the activity. Both the second and the third Cas7-like domains mediate target RNA cleavage in a metal-dependent manner. The structure and mutational data indicate that the long variable insertion to the fourth Cas7 domain has little impact on RNA processing or targeting, suggesting the possibility for engineering a compact and programmable RNA interference tool.

*For correspondence:
hong.li@fsu.edu

Competing interest: The authors declare that no competing interests exist.

## Editor's evaluation

This manuscript is a timely contribution to the CRISPR/Cas field: the mode of function of the type III-E Cas7-11 CRISPR-Cas system. This is an RNA-guided RNA targeting system only characterized last year. In contrast to Cas13 systems, Cas7-11 does not possess collateral damaging activity, hence does not show cytotoxicity when introduced into human cells. These are highly desirable traits in practical applications.

## Introduction

The CRISPR Cas systems confer adaptive immunity to prokaryotic hosts against invading viruses by encoding a range of different CRISPR Cas efforts that interfere with the invader nucleic acids (*Makarova et al., 2020*). Three types of CRISPR Cas effectors are known to utilize programmable CRISPR RNA (crRNA) to guide cleavage of the complementary target RNA. The multi-subunit Type III effectors, exemplified by the Type III-A (Csm) and the III-B subtypes (Cmr), assemble 4–5 Cas7, 2–3 Cas11, and 1 Cas10 subunits into a sea horse-shaped helical enzyme (*Molina et al., 2020*). They cleave the target RNA at a 6-nucleotide (nt) interval within the complementary region that coincides with the evenly spaced Cas7 subunits (*Elmore et al., 2016*; *Samai et al., 2015*; *Tamulaitis et al., 2014*). The Type VI, or Cas13, is a single subunit and substantially smaller effector. Unlike Csm/Cmr, Cas13 employs two higher eukaryotic

**eLife digest** Ribonucleic acid or RNA is an important molecule involved in making proteins, transmitting diseases, offering immunity in form of vaccines, and also degrading itself. Programmed RNA degradation is a common method used by bacteria to protect themselves from invading viruses.

Bacteria acquire viral genetic materials during infections, which are then converted into RNA fragments, or guide RNA. The guide RNAs both locate and recruit enzymes to help destroy the infectious RNA. These programmable RNA degradation machineries can be repurposed for biotechnology applications to help regulate gene expression or to minimize the effect of viral infections.

Similar machineries, like the CRISPR/Cas9 gene editing tool, act like genetic scissors, allowing researchers to make precise modifications to DNA to study and alter the role of genes in the cell. Like in bacteria, the CRISPR system uses fragments of RNA from viruses as a guide to identify matching targets and create breakages in the genetic material. Recently, researchers discovered Cas7-11, which is used to break sections of RNA in viruses.

To better understand how Cas7-11 works, Goswami et al. studied its three-dimensional structure. Detailed views of each segment of the protein, together with biochemical studies of the protein's activity, helped to identify their respective roles. The structural information also highlighted three regions involved in snipping RNA and revealed how they drive this process. This analysis showed that a short segment of Cas7-11 alone is sufficient to prepare and bind the guide RNA fragments.

These findings add to the understanding of how Cas7-11 prepares its guide and creates breakages in RNA. It has a similar structure to a previously known assembly of proteins that also breaks down RNA, providing insight into its evolution. The detailed analysis of how Cas7-11 works also demonstrates the possibility of engineering it as a laboratory tool to remove specific RNA sequences in cells.

and prokaryotic nucleotide binding domains in cleaving the target RNA outside the complementary region (*Abudayyeh et al., 2016*; *East-Seletsky et al., 2016*). The recently discovered Type III-E effector, Cas7-11 or gRAMP (for giant repeat-associated mysterious protein), is also a single subunit effector with fused Cas7-like and Cas11-like segments (*van Beljouw et al., 2021*; *Özcan et al., 2021*; *Figure 1a*). Unlike Cas13 but similar to Csm/Cmr, Cas7-11 employs the Cas7-like segments to cleave crRNA-guided target RNA (*Figure 1a*). Interestingly, whereas Cas13 can distinguish self from foreign RNA by utilizing the 3' protospacer flanking sequence (PFS) in the target RNA (*Meeske and Marraffini, 2018*), both Csm/Cmr and Cas7-11 are insensitive to 3' PFS in cleaving their respective target RNA (*van Beljouw et al., 2021*; *Özcan et al., 2021*). The three effectors also differ in crRNA processing. Csm/Cmr utilize an independent processing endonuclease, Cas6, to result in a mature crRNA containing an 8-nt repeat (5'-tag) linked to the spacer (*Carte et al., 2008*). By contrast, both Cas13 and Cas7-11 process their own crRNA (*East-Seletsky et al., 2016*; *van Beljouw et al., 2021*; *Özcan et al., 2021*). Cas7-11 is therefore believed to be a unique Type III CRIPSR-Cas system. Interestingly, Cas7-11 has been demonstrated to form a complex with the caspase-like TPR-CHAT peptidase, suggesting a potential for a viral RNA-induced and protease-mediated antiviral immunity (*van Beljouw et al., 2021*). Given the known collateral nuclease activities of Cas13 (*Abudayyeh et al., 2016*; *East-Seletsky et al., 2016*) and Csm/Cmr (*Elmore et al., 2016*; *Samai et al., 2015*; *Kazlauskiene et al., 2016*), and the complex enzyme composition of Csm/Cmr, Cas7-11 provides a desirable platform to further develop RNA interference and editing tools. To understand the molecular basis for crRNA processing and target interference by Cas7-11, we determined a cryo-electron microscopy (cryo-EM) structure of *Desulfonema ishimotonii* Cas7-11 (DiCas7-11) at an overall resolution of 2.82 Å (*Figure 1b-c*, *Figure 1—figure supplement 1*, *Figure 1—figure supplement 2*, *Supplementary file 1*). DiCas7-11 has been demonstrated to function in programmable RNA cleavage and editing both in vitro and in mammalian cells (*van Beljouw et al., 2021*; *Özcan et al., 2021*). Our structure provides the architecture of the enzyme and the molecular basis for its enzymatic activities.

## Results and Discussion

The wild-type DiCas7-11 was incubated with its precursor crRNA (pre-crRNA) and a complementary target RNA under a reactive condition before being made frozen specimen (*Figure 1—figure supplement 1*). Under this condition, DiCas7-11 successfully processes the pre-crRNA and cleaves the target RNA

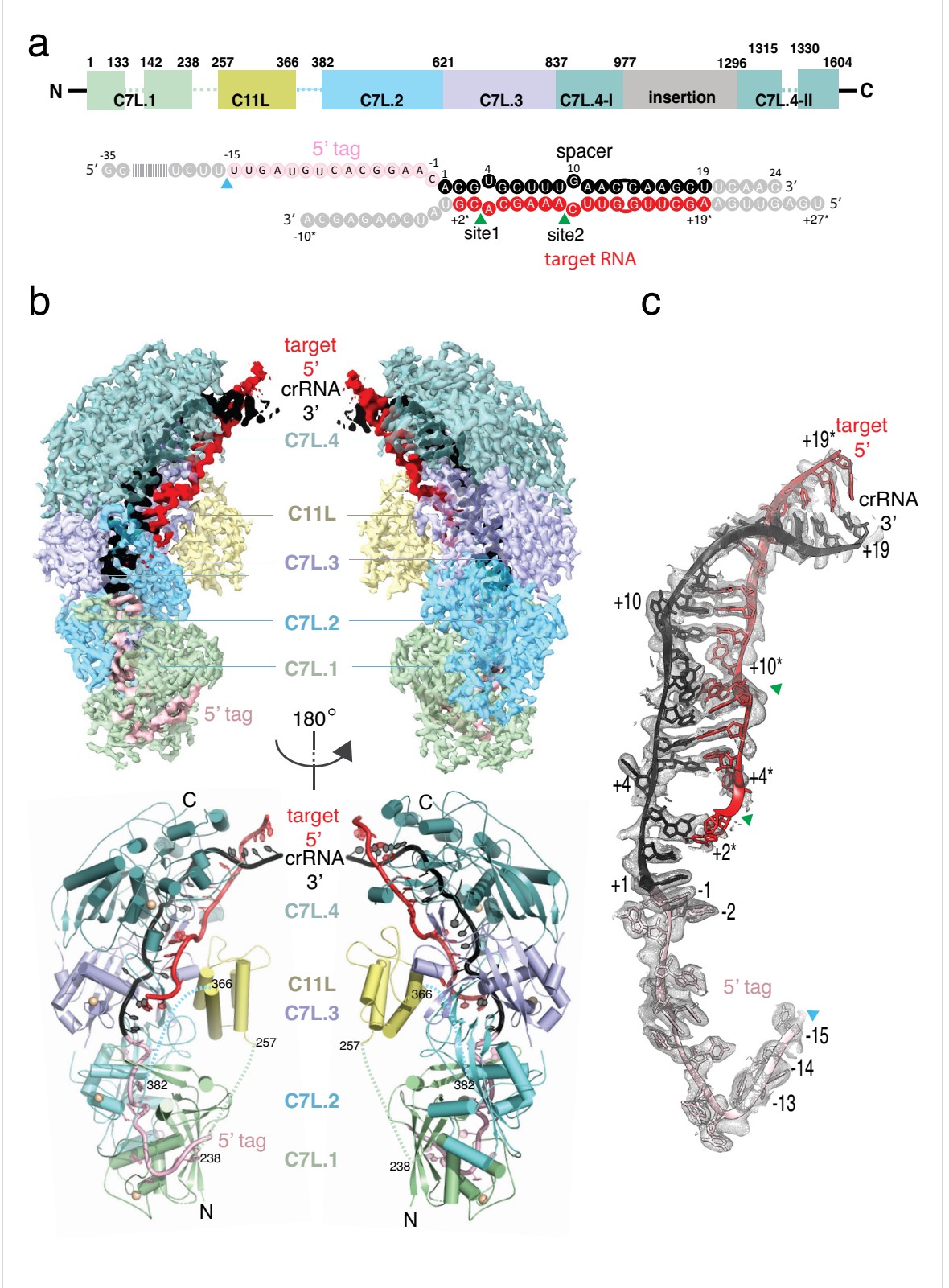

**Figure 1.** Structure overview of *Desulfonema ishimotonii* Cas7-11 (DiCas7-11)-crRNA-target RNA ternary complex. (**a**) Domain organization of DiCas7-11 and schematic representation of crRNA-target RNA duplexes used in the study. Protein domain and RNA elements are colored differently and labeled. C7L denotes Cas7-like domain and C11L denotes Cas11-like domain. The gray colored nucleotides indicate those included in the constructs but not built due to weak or no density. Dash lines indicate protein regions not built due to weak or no density. The cyan and green colored triangles indicate

*Figure 1 continued on next page*

*Figure 1 continued*

the precursor crRNA (pre-crRNA) processing and target RNA cleavage sites, respectively. (**b**) Top, electron potential density map of DiCas7-11-crRNA-target RNA ternary complex shown in two different orientations. Bottom, cartoon representation of DiCas7-11-crRNA-target RNA ternary complex shown the same views as in top panel with corresponding colors representing protein domains and the two RNA strands. The solid spheres in wheat represent Zn atoms. (**c**) Close-up view of the density for the crRNA (spacer, black; 5'-tag, light pink) and target RNA (red) duplex. RNA nucleotide positions as well as the three cutting sites are indicated as in panel (**a**).

The online version of this article includes the following source data and figure supplement(s) for figure 1:

**Figure supplement 1.** Top, schematic of the plasmid used for over-expression of the wild-type and mutant *Desulfonema ishimotonii* Cas7-11 (DiCas7-11), the elution profile of full-length DiCas7-11 on size exclusion chromatography, and gel analysis of the purified wild-type (DiCas7-11) and an insertion deleted Δint1-Cas7-11 proteins.

**Figure supplement 1—source data 1.** Polyacrylamide gel images showing Cas7-11 purification and Cas7-11-crRNA complex formation used in *Figure 1—figure supplement 1*.

**Figure supplement 2.** Data collection, processing,and assesment.

**Figure supplement 3.** Close-up views of electron potential density maps for selected regions and comparison of the target:crRNA duplex to standard A form helix.

**Figure supplement 4.** Comparison of *Desulfonema ishimotonii* Cas7-11 (DiCas7-11) to the homologous multi-subunit Csm complex from *Lactococcus lactis* (LlCsm) (PDB ID: 6XN2) (**a and b**) and its individual protein domains (**d**).

**Figure supplement 5.** Schematic of RNA-protein interactions observed in the *Desulfonema ishimotonii* Cas7-11 (DiCas7-11)-crRNA-target RNA ternary complex structure.

(*Figure 2* and *Figure 1—figure supplement 1*). The density map resolves most of the DiCas7-11 protein, the crRNA (–15 to +19) and the partially cleaved target RNA (+19* to +2*) (*Figure 1b-c*, *Figure 1—figure supplement 2*, and *Figure 1—figure supplement 3*). The PFS region plus the first base paired nucleotide (+1*) are not observed. The core Cas7-11 assembly is half-moon shaped with four Cas7-like (C7L) domains (C7L.1–C7L.4 from N- to the C-terminus) forming a long ridge and a single Cas11-like domain (C11L) occupying the crescent center. The entire DiCas7-11 complex can be superimposed onto the closely matched homologous *Lactococcus lactis* Csm (LlCsm) complex (*Sridhara et al., 2022*) (Type III-A) (*Figure 1—figure supplement 4a*) with the C7L.1-C7L.4 ridge matching that formed by four Cas7-like (Csm3) subunits and C11L matching the Cas11-like (Csm2) subunit adjacent to Cas10 (Csm1) (*Figure 1—figure supplement 4b*). The similarity in the bound RNA trajectory at the core between the two complex is striking (*Figure 1—figure supplement 4b*), suggesting a similar RNA binding apparatus in both complexes. DiCas7-11 lacks the domains equivalent to the Csm4 and the Cas10 (Csm1) subunits and, thus, the functions associated with them. In Csm complexes, Csm4 recognizes and stabilizes the 8-nt 5'-tag derived from the direct repeat whereas Cas10 carries out cyclic oligoadenylate synthesis and ancillary DNA cleavage (*Sridhara et al., 2022*; *You et al., 2019*; *Jia et al., 2019*). Cas10 also secures the 3' PFS of the cognate target RNA, and thus, plays a role in discrimination of self from foreign RNA (*Sridhara et al., 2022*; *You et al., 2019*; *Jia et al., 2019*).

Despite the overall structural similarity between the C7L- and the LlCsm-formed ridge, each C7L differs slightly in protein sequence and folding (*Figure 1—figure supplement 4c*), which gives rise to their different roles in binding and cleaving RNA. The 34-nt crRNA lies along the C7L ridge with its repeat region spanning C7L.1-C7L.2 and the spacer region covering C7L.3-C7L.4 (*Figure 1b-c* and *Figure 1—figure supplement 3*). The first 18 nucleotides of the target RNA (+19* to +2*) remain base paired with the spacer region of the crRNA (*Figure 1c*, *Figure 1—figure supplement 3*, and *Figure 1—figure supplement 5*). The short span of the guide-target region on C7L domains explains the two, instead of four as in LlCsm, sites of target cleavage (*van Beljouw et al., 2021*; *Özcan et al., 2021*).

The observed structure suggests that the pre-crRNA is processed by the first C7L domain, which yields a mature crRNA containing the last 15 nucleotides of the direct repeat (5'-tag) linked to the programmed spacer (*Figure 1c*, *Figure 2a*, and *Figure 1—figure supplement 3*). To confirm the site of processing, we subjected synthetic pre-crRNA containing 2'-deoxy modification at −16,–15, or −14 position to the processing reaction, respectively, and found that the cleavage products are consistent with the density-derived 15-nt 5'-tag (*Figure 2b* and *Figure 1—figure supplement 3*). Strikingly, the sequence identity upstream of U(–15) is not important for processing, as those substituted with poly-adenine (polyA) or the previously characterized *Candidatus Scalindua broadae* (Csb) pre-crRNA are successfully processed by DiCas7-11 (*Figure 2c*). A well-conserved histidine residue, His43, is immediately next to the leaving

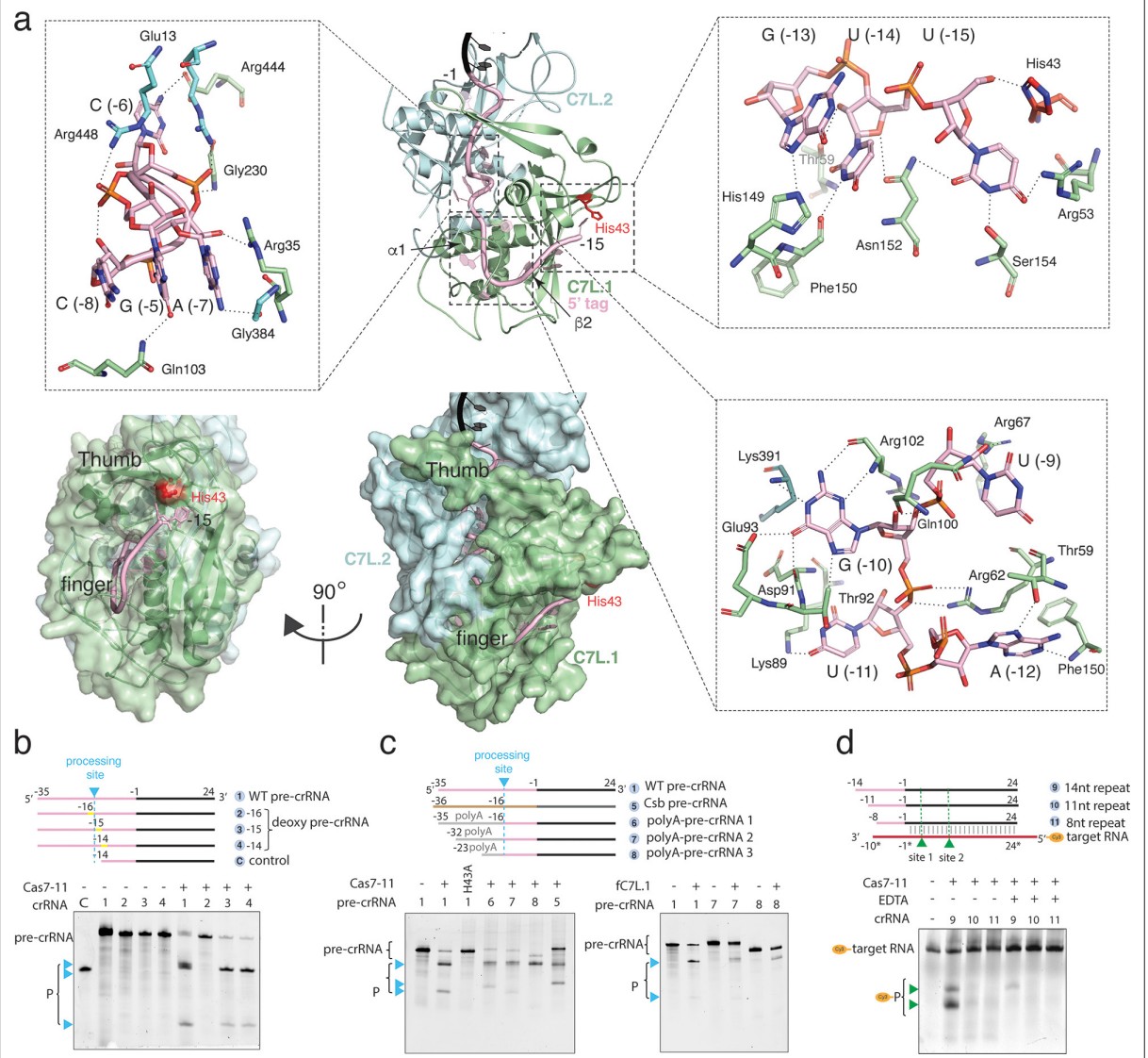

**Figure 2.** Precursor crRNA (pre-crRNA) processing and recognition. (**a**) The mode of Cas7-like domain 1 (C7L.1) and C7L.2 interaction with the processed crRNA nucleotides –15 to –1 in both cartoon (top) and surface (bottom) representations. Key secondary elements involved in crRNA interaction are labeled. Insets indicate close-up views around U(–15)-U(–13)-G(–13), the C(–8)-A(–7)-C(–6)-G(–5) tight RNA turn, and the conserved A(–12)-U(–11)-G(–10)-U(–9) tetranucleotide. The catalytic residue His43 for crRNA processing is colored red. Dash lines indicate close polar contacts. (**b–c**) Top, various pre-crRNA used in processing reactions. Cyan colored triangles and dash lines indicate the pre-crRNA processing sites. Yellow bars indicate the sites of deoxy modification. The control RNA contains the last 14 nucleotides of the repeat plus the spacer. Spacer and repeat are shown in black and pink, respectively. 'Csb pre-crRNA' denotes the pre-crRNA for *Candidatus Scalindua broadae* Cas7-11. Processed products (P) of pre-crRNA are stained by SYBR Gold and imaged by ChemiDoc MP. Bottom, RNA processing results analyzed on polyacrylamide urea gel for the wild-type (1) and other pre-crRNA (2-8) by the wild-type *Desulfonema ishimotonii* Cas7-11 (DiCas7-11) (WT), the His43 to alanine mutant (H43A) of DiCas7-11, and the free-standing C7L.1 (fC7L.1). Processing products are indicated by cyan triangles. (**d**) Target RNA cleavage results analyzed on polyacrylamide urea gel using the wild-type and truncated pre-crRNA in the presence and absence of ethylenediaminetetraacetic acid (EDTA). 'Cy3' denotes the target RNA containing a 5'-Cy3 fluorophore. The cleavage products (P) of the Cy3-labeled target RNA are visualized on ChemiDoc MP using 550 nm as the excitation and 564 nm as the emission wavelength, respectively, and are indicated by green triangles.

The online version of this article includes the following source data and figure supplement(s) for figure 2:

**Source data 1.** Polyacrylamide gel image for deoxy precursor crRNA (pre-crRNA) processing activity shown in *Figure 2*.

**Source data 2.** Polyacrylamide gel image for precursor crRNA (pre-crRNA) variant processing by DiCas7-11 and free-standing C7L.1 (fC7L.1) shown in *Figure 2*.

**Source data 3.** Polyacrylamide gel image of target RNA cleavage activities with truncated crRNA and DiCas7-11 shown in *Figure 2*.

*Figure 2 continued*

**Figure supplement 1.** Sequence comparison among Cas7-11 proteins overlays with the secondary structure of DiCas7.

**Figure supplement 2.** Protein analysis and electrophoretic mobility shift assay of the free-standing C7L.1 (fC7L.1) with precursor crRNA (pre-crRNAs).

**Figure supplement 2—source data 1.** Native-polyacrylamide gel images for binding activity by free-standing C7L.1 (fC7L.1) used in *Figure 2—figure supplement 2*.

**Figure supplement 3.** Comparison of the catalytically inactive (PDB: 7WAH) (gray) and the catalytically active (PDB: 8D1V) (color) *Desulfonema ishimotonii* Cas7-11 (DiCas7-11)-RNA complex structures.

5′-hydroxyl oxygen (2.3 Å) of U(–15), suggesting its role in processing. Consistently, His43 to alanine mutation (H43A) abolished pre-crRNA cleavage (*Figure 2c*). The requirement for His43 without requiring divalent metals in processing suggests that DiCas7-11 employs a general acid-base catalysis mechanism similarly suggested for Cas6 (*Li, 2015*; *Hochstrasser and Doudna, 2015*), although detailed roles of His43 and other possible catalytic residues remain to be characterized. In this regard, the histidine-mediated processing reaction may be further assisted by nearby polar residues Tyr55 (4.3 Å from the leaving 5′-hydroxyl oxygen) and Arg26 (5.1 Å from the leaving 5′-hydroxyl oxygen). Interestingly, CsbCas7-11 contains threonine in place of His43 and phenylalanine in place of Tyr55, and it does not process its crRNA at position –15 (*van Beljouw et al., 2021*), suggesting that precise processing may not be required for the RNA interference activity.

The processed 15-nt 5′-tag interacts extensively with C7L.1 and to less extent with C7L.2 (*Figure 2a* and *Figure 1—figure supplement 5*). The protein nearly buries the entire 5′-tag and thus precluded it from complete base pairing with a complementary RNA. In general, well-conserved residues form base-specific interactions while non-conserved residues maintain interactions with RNA sugar phosphate backbone. C7L.1 forms a ferredoxin fold (β2↑β3↓α2β1↑α1β4↓) highly abundant in CRISPR Cas components (*Makarova et al., 2011*; *Wang and Li, 2012*). It uses α1, β2, a long β-hairpin connecting β2 to β3 (thumb) and a C7L.1-specific insertion loop (finger) to secure the significantly bent 5′-tag, resembling a rope-gripping right hand (*Figure 2a* and *Figure 1—figure supplement 5*). The first nucleotide, U(–15), forms a hydrogen bond network with a number of residues including the well-conserved His43 (*Figure 2a*, *Figure 1—figure supplement 3*, and *Figure 1—figure supplement 5*). Removal of U(–15) did not impact target RNA cleavage (*Figure 2d*), suggesting the possibility that the U(–15)-protein interactions may be important for processing rather than target interference. The most extensive interactions take place at the strictly conserved A(–12)-U(–11)-G(–10)-U(–9) tetranucleotide (*Figure 2a*). Both sidechain as well as mainchain atoms of C7L.1 participate in 'reading' the four RNA bases, and the amino acids involving base-specific contacts (Thr59, Arg62, Arg67, Lys89, Thr92, and Glu93) are well conserved (*Figure 2—figure supplement 1*). Strikingly, all edges of G(–10), the Watson-Crick, the Hoogsteen, and the sugar, are in close contacts with the conserved C7L.1 residues (*Figure 2a*, *Figure 1—figure supplement 5*, and *Figure 2—figure supplement 1*). Downstream of the AUGU tetranucleotide is a tight right-handed helical turn formed by C(–8)-A(–7)-C(–6)-G(–5). The turn is stabilized by both base stacking as well as an unusual network of intra-strand polar contacts. The base of G(–5) interdigitates those of C(–8) and A(–7) with the well-conserved Arg35 on top, leaving C(–6) protruding into the interior of the protein (*Figure 2a*, *Figure 1—figure supplement 5*, and *Figure 2—figure supplement 1*). Whereas phosphate backbone atoms in A form RNA do not engage in intra-strand contacts, those within the turn mediate numerous interactions (*Figure 2a*). The strictly conserved G(–5) forms the most intra-strand interactions. Its N2 atom contacts the non-bridging oxygen of A(–7) while its non-bridging oxygen forms hydrogen bond with 2′-OH of C(–8) (*Figure 2a*). Finally, the 2′-OH of G(–5) forms bifurcated contacts with the N7 atom of the strictly conserved A(–7) and G(–4). The tight and the N+3 interactions involving the C(–8)-A(–7)-C(–6)-G(–5) backbone atoms is reminiscent of those within a $3_{10}$ helix in proteins. Interestingly, the non-conserved 5′-tag nucleotides, A(–3), C(–6), and C(–8) form minimal contacts with the protein, suggesting their minor roles in maintaining protein-RNA interactions (*Figure 2a*, *Figure 1—figure supplement 5*, and *Figure 2—figure supplement 1*). The rest of the 5′-tag is clamped down by the thumb of C7L.1 and α1 of C7L.2 analogously as by two Cas7 subunits in Csm complexes (*Sridhara et al., 2022*). Consistent with the extensive 5′-tag-protein interactions, removal of either first four or eight 5′-tag nucleotides abolished RNA-guided target cleavage (*Figure 2d*). The different mode of 5′-tag binding to Cas7-11 from that of Csm explains the lack of a Cas10-like effector as in Csm. Rather, it allosterically activates a different effector, the TPR-CHAT protease, in a 3′ PFS-dependent manner. Though 13 of the 15 5′-tag nucleotides

are buried by the protein, the last two, C(–1) and A(–2), are partially solvent-accessible and can potentially participate in PFS-mediated activation of the TRP-CHAT activity (*Hu et al., 2022*).

Based on structural observations, we hypothesized that a free-standing C7L.1 (fC7L.1) domain may be sufficient for binding and processing pre-crRNA. We thus created and purified a truncated mutant comprised of residues 1–238 (hereafter fC7L.1 for free C7L.1) (*Figure 2—figure supplement 2a*). Following 30 min incubation with pre-cRNA, fC7L.1 successfully processed the RNA, albeit at slightly reduced efficiency as compared to wild-type DiCas7-11 (*Figure 2c*, right). In addition, electrophoretic mobility shift assay shows that fC7L.1 remains bound to the cleaved pre-crRNA specifically (*Figure 2—figure supplement 2b*), suggesting its potential to be used as a new CRISPR tool for in vivo RNA tracking or purification.

The spacer region of the crRNA is captured by the rest of the C7L ridge and base paired with the target RNA (*Figure 3a*). The three base paired regions largely resemble A form helix with some deviations in base-pair tilt (*Figure 1—figure supplement 3b*). Three of the four C7L domains contain the characteristic 'thumb' that secure the crRNA at two evenly spaced (6-nt) kinks with bases flipped. The duplex bound by C7L.4 creates an extra spacing between the G(+13*)-C(+13) and G(+14*)-C(+14) pairs but no base flipping. Like the multi-subunit Type III effectors, the kinked crRNA-target RNA duplex creates bended sugar-phosphate backbone at the locations that coincide with the sites of cleavage. Both sites are cleaved by Cas7-11 in a metal-dependent manner (*Figure 3b and c*). Consistent with the previous observation that Type III systems generate 2',3'-cyclic product in spite of the metal dependence (*Hale et al., 2009*), the target RNA containing 2'-deoxy modification at A(+4*) (site 1) and C (+10*) (site 2) prevented formation of any cleavage product (*Figure 3b and d*). Two acidic residues, Asp429 and Asp654, were previously shown to be critical to cleavage at site 1 and site 2, respectively (*Özcan et al., 2021*). Satisfactorily, they are found near each corresponding scissile phosphate with the carboxylate oxygen 4.2–6.7 Å from the leaving 5'-oxygen (*Figure 3a*). At both sites, the phosphodiester bond breakage is further assisted by the near 'in-line' geometry of the nucleophilic 2'-oxygen, the scissile phosphate and the leaving 5'-oxygen (*Figure 3a*). Two residues from the C11L domain, Arg283 (site 1) and His306 (site 2), where Arg283 is better conserved than His306, are observed to stabilize the attacking nucleotides by stacking on their bases (*Figure 3a*). Despite the demonstrated dependence on $Mg^{2+}$ in target cleavage (*Figure 3c*), no sufficient density is observed that can be assigned to $Mg^{2+}$ ions. The roles of these protein residues in facilitating metal-dependent phosphodiester bond breakage remains unclear.

Interestingly in the homologous CsbCas7-11, Asp429 is not conserved and mutation of its equivalent Asp448 and other surrounding residues did not impact site 1 cleavage (*van Beljouw et al., 2021*). To access possible roles of other residues near site 1 in catalysis, we mutated the well-conserved Tyr360 of C11L given its proximity to A(+4*) (*Figure 3a*). Surprisingly, we found that Tyr360 is not required for site 1 cleavage (*Figure 3b and f*), indicating that, at least for DiCas7-11, Asp429 is sufficient in mediating the metal-dependent cleavage.

To access the length of base pairing required for target cleavage, we further examined the cleavage of a series of truncated target from either the 3' or the 5' end. We found that as short of 16 base pairs in total length and 2 base pairs on one flanking end can facilitate RNA cleavage (*Figure 3b and e*), suggesting that both protein and crRNA:target pairing play significant roles in shaping the target RNA for cleavage.

Though the final atomic model of DiCas7-11 lacks the large insertion to C7L.4 (residues 979–1297) due to weak density, focused classification and refinement led to a low-resolution map that matches the AlphaFold-predicted model of the insertion domain (*Figure 4a and b*). This model indicates that a large majority of the insertion domain is not engaged with any of the features described above and suggests the possibility that it is not essential to RNA-guided target cleavage. To test this hypothesis, we removed residues 1009–1220 to create DiCas7-11-Δint1. Consistently, we showed that DiCas7-11-Δint1 retains almost all RNA-guided target cleavage in an in vitro assay (*Figure 4c*).

A structure of catalytically inactive DiCas7-11 (Asp429 and Asp654 to alanine) bound with a slightly modified crRNA (U(–15) is replaced by G(–15)) recently became available (PDB ID: 7WAH) (*Kato et al., 2022*). We superimposed the active form of DiCas7-11 bound with the processed native crRNA (this study) to that by Kato et al. (*Figure 2—figure supplement 3*) and found a close agreement in protein and RNA structures. The superimposed structures resulted in 0.918 Å root-means-square-difference (r.m.s.d.) for 8114 atoms for protein, 0.691 Å for 192 backbone atoms of crRNA, and 0.80 Å for 96 backbone atoms of target, respectively (*Figure 2—figure supplement 3a*). The target cleavage sites are highly similar despite the protein used by *Kato et al., 2022*, contains mutated catalytic aspartates. The largest

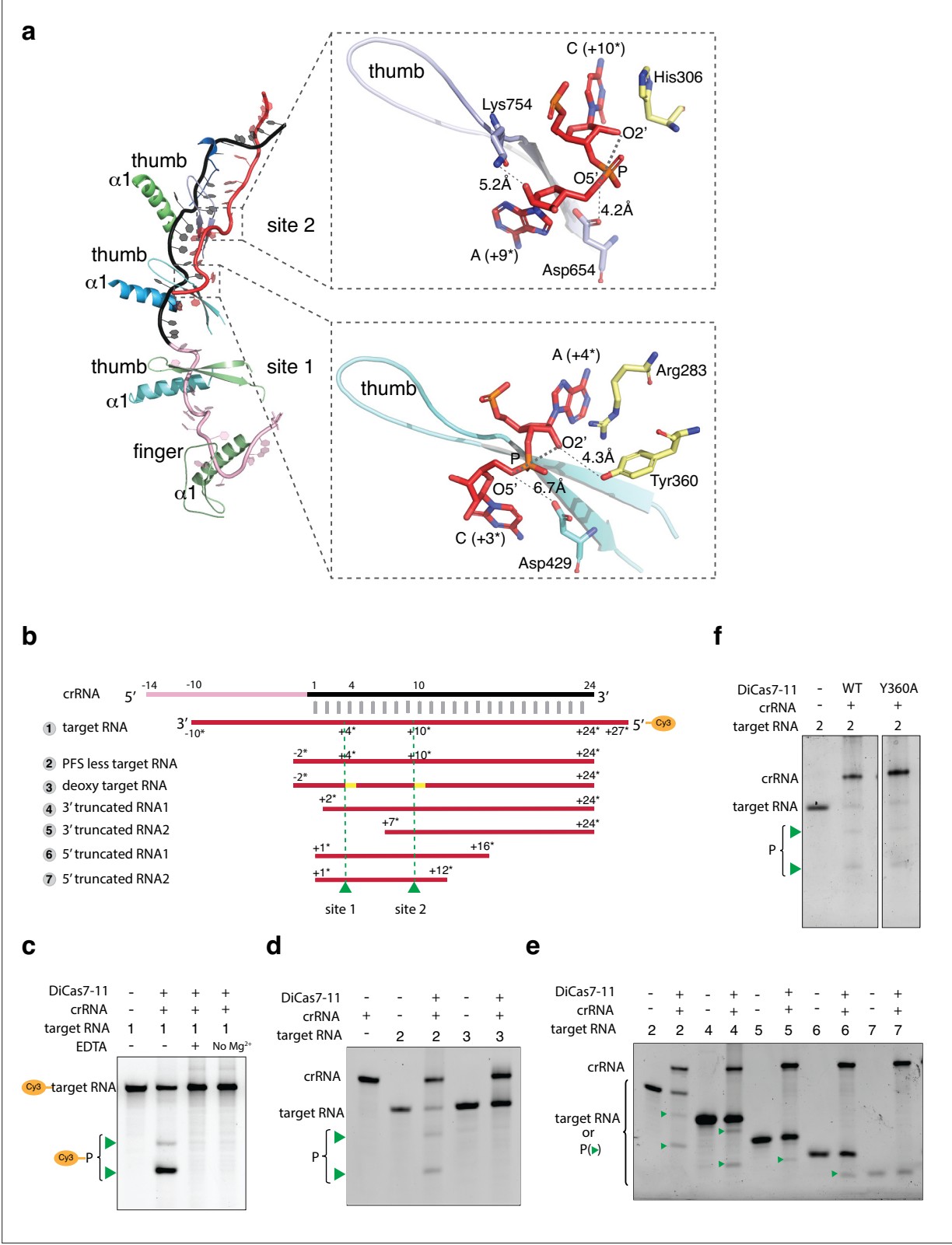

**Figure 3.** Target RNA cleavage mechanism. (**a**) Recognition of target RNA by the crRNA and *Desulfonema ishimotonii* Cas7-11 (DiCas7-11). The ferredoxin fold α1 and the thumb hairpin for each of the four Cas7-like (C7L) domains are shown as cartoons and colored as in **Figure 1**. The quoted 'thumb' indicates the degenerate thumb motif for the C7L.4 domain. Insets show the two target cleavage sites in close-up views. RNA nucleotides and key amino acids are shown in stick models. The three atoms involved in formation of the 'in-line' geometry during phosphodiester bond breakage

*Figure 3 continued on next page*

*Figure 3 continued*

are labeled and indicated by thick dash lines. The closest of the three atoms to the putative catalytic residues, Asp654 (for site 2) and Asp429 (for site 1), are indicated by a connecting dash line. The close contact between Tyr360 and A(+4*) 2'-hydroxyl oxygen at site 1 and that between Lys754 and A(+9*) 2'-hydroxyl at site 2 are also indicated by dash lines. (**b**) Schematic of the Cy3-labeled (substrate 1) and other target RNA (substrates 2–7) used in cleavage activity assays. Yellow bars mark the locations of the deoxy modification on the target RNA substrate 3. Green colored triangles and dash lines indicate target RNA cleavage sites. (**c–f**) Target RNA cleavage by DiCas7-11 and its Tyr360 to alanine mutant (Y360A) are analyzed on polyacrylamide urea gel. The cleavage products (P) of the Cy3-labeled target RNA are visualized on ChemiDoc MP using 550 nm as the excitation and 564 nm as the emission wavelength, respectively. Cleavage products (P) of non-Cy3-labeled target RNA are stained by SYBR Gold and imaged by ChemiDoc MP. All cleavage products are indicated by green triangles.

The online version of this article includes the following source data for figure 3:

**Source data 1.** Polyacrylamide gel image of target RNA cleavage activity by DiCas7-11 in presence and absence of metal ions shown in *Figure 2*.

**Source data 2.** Polyacrylamide gel image of deoxy-target RNA cleavage activity by DiCas7-11 shown in *Figure 3*.

**Source data 3.** Polyacrylamide gel image of target RNA variant cleavage activity by DiCas7-11 shown in *Figure 3*.

**Source data 4.** Polyacrylamide gel image of target RNA cleavage activity by *Desulfonema ishimotonii* Cas7-11 (DiCas7-11) Y360A mutant shown in *Figure 3*.

difference, however, is observed in interactions between DiCas7-11 and the first 5'-tag residue between the two structures. In that by Kato et al., G(–15), which is non-native, establishes minimal interactions with the protein, whereas in that of the active DiCas7-11, the native U(–15) interacts extensively with conserved protein residues (*Figure 2a*, *Figure 1—figure supplement 3*, and *Figure 2—figure supplement 3b*). In addition, whereas catalytic His43 forms a close contact with the 5'-OH leaving group of U(–15), it is at a greater distance to G(–15) (*Figure 2—figure supplement 3b*). This result suggests that, though U(–15) is nonessential to the architecture and the target-cleavage competency of DiCas7-11, it may be important for efficient processing.

The structure and complementary biochemical assays show that Cas7-11 has a minimal architecture required for programmable RNA cleavage. The covalent linkage of the homologous units suggests an evolutionary advantage in dedicating Cas7-11 to RNA cleavage. Considering the known collateral RNase activity of Cas13 and the complicated Csm/Cmr systems, Cas7-11 offers a desirable alternative in developing gene regulation tools. While DiCas7-11 has been successfully demonstrated to function in mammalian cells, the efficiency and accuracy remain to be improved. With now the available structures and the accurately mapped processing and target cleavage sites, protein engineering may assist the efforts in designing improved Cas7-11-derived RNA interference platforms. Both fC7L.1 and DiCas7-11-Δint1 lend a proof-of-concept for such an effort.

# Materials and methods

## Key resources table

| Reagent type (species) or resource | Designation | Source or reference | Identifiers | Additional information |
|---|---|---|---|---|
| Strain, strain background (species) | *Escherichia coli* NiCo21(DE3) | New England Biolabs | C2529H | Used for proteins expression |
| Recombinant DNA reagent | Sumo-tag-DiCas7-11 expression plasmid | *Özcan et al., 2021* | Addgene:172503 | |
| Recombinant DNA reagent | His-tag-DiCas7-11 expression plasmid | This paper | N/A | Constructed in-house. Protein-encoding sequences are inserted into pACYC-duet-1 cloning vector with *Bam*HI and *Eco*RI cut sites, T7 promoter, p15A origin, and chloramphenicol resistance |
| Recombinant DNA reagent | Cas7-11-Δint-1 expression plasmid | This paper | N/A | Addgene:172503; see *Supplementary file 2* for primers |
| Recombinant DNA reagent | fC7L.1 expression plasmid | This paper | N/A | |

*Continued on next page*

*Continued*

| Reagent type (species) or resource | Designation | Source or reference | Identifiers | Additional information |
|---|---|---|---|---|
| Peptide, recombinant protein | Cas7-11- Δint-1 protein | This paper | N/A | |
| Peptide, recombinant protein | fC7L.1 protein | This paper | N/A | Expressed and purified in house from *E. coli* NiCo21(DE3) cells |
| Peptide, recombinant protein | ULP1 protease | Protein Expression Facility, FSU | N/A | |
| Software, algorithm | cryoSPARC (v3.3.1) | *Punjani et al., 2017* | https://cryosparc.com | |
| Software, algorithm | RELION 4.0 | *Kimanius et al., 2021* | https://www2.mrc-lmb.cam.ac.uk/ | |
| Software, algorithm | COOT | *Emsley et al., 2010* | https://www2.mrc-lmb.cam.ac.uk/personal/pemsley/coot/ | |
| Software, algorithm | PHENIX | *Afonine et al., 2018* | https://phenix-online.org | |
| Software, algorithm | UCSF ChimeraX UCSF Chimera | *Pettersen et al., 2021 Pettersen et al., 2004* | https://www.rbvi.ucsf.edu/chimera/ | |

## Protein expression and purification

Two DiCas7-11 expression constructs were tested for protein expression. The first one comprises pACYC-duet-1 plasmid inserted with the codon-optimized sequence encoding for DiCas7-11 with an N-terminal his-tag. The second plasmid encodes sumo-tag fused DiCas7-11 (Addgene: 172503). *Escherichia coli* NiCo21 (DE3) competent cells were transformed with either plasmid used in protein production. A single colony was picked and transferred to 100 mL LB media containing 50 µg/mL ampicillin and grown for 12 hr at 37°C before inoculation into 1 L LB culture. The cells were induced at mid-log phase with the addition of 0.5 mM IPTG (isopropyl-β-D-thiogalactopyranoside) and grown overnight at 16°C and harvested. Cells were lysed and centrifuged at 4000 rpm for 30 min in buffer A (20 mM Tris pH 8.0, 500 mM NaCl, 5 mM β-mercaptoethanol, and 5% glycerol) sonicated 10 times on pulse for 20 s with 40 s rest between the pulses. The cell lysate was centrifuged at 16,000 rpm for 1 hr at 4°C and the resulting supernatant was passed through the pre-equilibrated Ni-NTA resin column. The protein bound resins were washed by 100 mL buffer B (20 mM Tris pH 8.0, 500 mM NaCl, 5 mM β-mercaptoethanol, 5% glycerol, and 50 mM imidazole) and eluted with buffer C (20 mM Tris pH 8.0, 500 mM NaCl, 5 mM β-mercaptoethanol, 5% glycerol, and 300 mM imidazole). For the sumo-tagged DiCas7-11, the ULP1 protease was added to the elutant to remove the sumo-tag from DiCas7-11 while dialyzing at 4°C overnight. The digested protein solution was diluted twofold before being loaded onto a heparin column pre-equilibrated with buffer D (20 mM Tris pH 8.0, 250 mM NaCl, 5 mM β-mercaptoethanol, and 5% glycerol). The bound protein was eluted with a salt gradient. Pooled fractions were further purified on a gel filtration column in buffer E (20 mM Tris pH 8.0, 500 mM NaCl, 2 mM DTT, and 5% glycerol). The protein containing fractions were pooled, concentrated to 21 mg/mL, aliquoted, and stored at –80°C for future use. We found that the quality of the sumo-tag purified protein is superior to that of the his-tag purified protein and used it in subsequent experiments. The DiCas7-11 mutants, DiCas7-11-Δint1 and fC7L.1, were prepared by Q5 ×2 master mix mutagenesis kit (New England Biolabs) using primers listed in *Supplementary file 2* and purified similarly as the wild-type sumo-tagged DiCas7-11.

## In vitro transcription and purification

For synthesis of 59-nt pre-crRNA, DNA oligonucleotides appended with T7 promoter sequence were ordered from Eurofins (*Supplementary file 2*). The complementary oligos, 50 µM in concentration, were annealed at 95°C followed by gradual cooling to 25°C at 1°C per minute rate. Next, 2.5 µL of annealing reaction was mixed with a transcription reaction master mix 50 mM Tris pH 8.0, 10 mM DTT 20 mM $MgCl_2$, 0.5 mM NTPs, and 48 µg/mL T7 RNA polymerase in a 50 µL reaction. The in vitro transcription reaction was kept overnight at 37°C and treated with 2 U of Turbo DNase (Invitrogen) for 1 hr at 37°C. The final product was purified by Monarch RNA Cleanup kit (New England Biolabs), eluted in water, flash-frozen using liquid nitrogen and stored at –80°C.

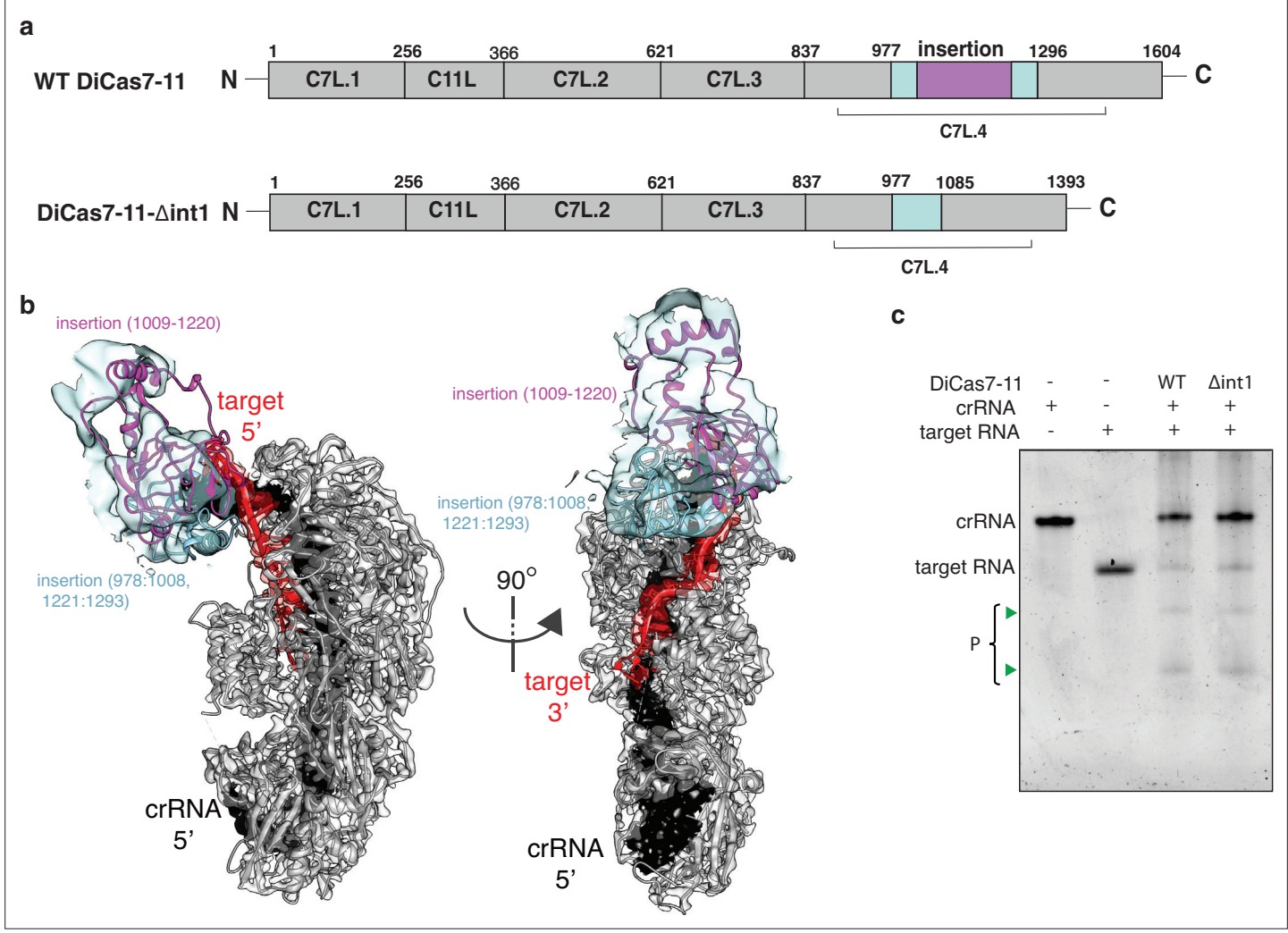

**Figure 4.** Engineering a compact *Desulfonema ishimotonii* Cas7-11 (DiCas7-11). (**a**) Schematic of domain organization of wild-type and an insertion deletion variant DiCas7-11-Δint1. The region removed is colored in purple and numbered. (**b**) Cartoon representation of DiCas7-11 overlaying with density map resulted from focused classification using a mask around the insertion domain. The insertion structure model is from AlphaFold prediction. (**c**) Target RNA cleavage by DiCas7-11 (WT) and DiCas7-11-Δint1(Δint1) are analyzed on a polyacrylamide urea gel. Cleavage products (P) are stained by SYBR Gold and imaged by ChemiDoc MP and are indicated by green triangles.

The online version of this article includes the following source data for figure 4:

**Source data 1.** Polyacrylamide gel image showing target RNA cleavage activity by DiCas7-11-Δint1.

## Pre-crRNA processing

Five hundred nM of DiCas7-11 of fC7L.1 was incubated with 500 nM of pre-crRNA at 37°C for 30 min in a 15 μL reactions containing ×1 processing buffer (40 mM Tris, pH 8.0, 70 mM NaCl). The reactions were stopped by using ×2 formamide dye (95% formamide, 0.025% SDS, 0.025% xylene cyanol FF, 0.5 mM EDTA). The samples were heated at 95°C for 5 min and run on 15% TBE-UREA gels (EC6885BOX, Thermo Fisher Scientific) at RT at 180 V in ×1 Tris Borate EDTA (TBE) running buffer. The gels were stained with SYBR Gold II (Invitrogen) stain and scanned by BIO-RAD ChemiDoc MP scanner.

## Target RNA cleavage

The in vitro target RNA cleavage assays were performed in a cleavage buffer containing 40 mM Tris pH 8.0, 70 mM sodium chloride, 10 mM $MgCl_2$. A binary complex was first prepared by incubating 400 nM Cas7-11 with 500 nM crRNA for 30 min at 37°C. The resulting samples were then incubated

with 500 nM target RNA (5Cy3 labeled or non-labeled) for 1 hr at 37°C. To test the metal dependence cleavage activity, reaction was supplemented with 16 mM EDTA. The reactions were stopped by using ×2 formamide dye (95% formamide, 0.025% SDS, 0.025% xylene cyanol FF, 0.5 mM EDTA). The samples were heated at 95°C for 5 min and run on 15% TBE-UREA gels (EC6885BOX, Thermo Fisher Scientific) at RT at 180 V in ×1 TBE running buffer. The gels were stained with SYBR Gold II (Invitrogen) stain and scanned by Bio-Rad ChemiDoc MP scanner. When Cy3-labeled RNA was used, gels were directly scanned by Bio-Rad ChemiDoc MP scanner.

### Electrophoretic mobility shift assay

For electrophoretic mobility shift assay, 1 μM fC7L.1 was separately added with 150 nM of polyA pre-crRNA (pA3) and non-specific RNA in the buffer containing 20 mM Tris pH 7.5, 70 mM NaCl, 5% glycerol, and 25 μg/mL heparin. The reaction was incubated for 30 min at 37°C. After the reaction was complete, the samples were mixed with ×6 loading dye (B7025S, New England Biolabs) and run on a 10% TBE gel (EC62752BOX, Thermo Fisher Scientific) at 4°C at 150 V. Gels were stained with SYBR Gold II (Invitrogen) and scanned by Bio-Rad ChemiDoc MP scanner.

### Sample preparation and data collection for cryo-EM studies

To reconstitute the ternary complex, 400 μg wild-type DiCas7-11 protein was incubated with 1.5 molar excess of a pre-crRNA in the buffer (30 mM Tris pH 8.0, 60 mM NaCl) at 37°C for 45 min followed by separation on a Superdex 200 increase 10/300 gel filtration column in the buffer (30 mM HEPES pH 7.5, 180 mM NaCl, 10 mM $MgCl_2$, and 2 mM TCEP). The peak fraction at 0.3 mg/mL determined by UV 280 nm absorbance was collected and further incubated with 2 molar excess target RNA at 37°C for 10 min before grid preparation. The Cas7-11-crRNA-target RNA sample in 5 μL volume was added onto glow-discharged 300 mesh Cu R1.2/1.3 holey carbon grids (Quantifoil) with extra layer of carbon (2 nm), blotted for 3 s at 100% humidity using FEI Vitrobot Mark IV. After flash-freezing in liquid ethane, the grids were transferred to liquid nitrogen for storage until cryo-EM imaging.

The Cas7-11 ternary complex micrographs were collected using EPU software on the Krios G3i cryo TEM (Thermo Fisher Scientific) equipped with Gatan Bioquantum K3 direct electron detector (Gatan) with 15 eV energy filter in a counted super-resolution mode. All 4177 images were collected at a dose rate of 60 e⁻/Å² with 1e⁻/Å² per frame in a pixel size of 0.825 Å/pixel. Motion correction was performed in bin 2 using MotionCorr 2 (*Zheng et al., 2017*) in a wrapper provided in Relion 4.0 (*Kimanius et al., 2021*) and contrast transfer function parameters were estimated with Gctf (*Zhang, 2016*) implemented in cryo-SPARC (*Punjani et al., 2017*). The stack was generated and imported to cryoSPARC for particle picking and 2D classification. The images with bad ice, astigmatism, drift, and poor sample quality were rejected resulting in 4168 images for further processing and particle picking, which resulted in a total of 1,301,452 particles. Several rounds of 2D classification led to 645,053 particles with good quality. RELION-4.0 was used to classify the particles, which led to further reduction of particles to 226,320 based on high-resolution features for reconstruction.

### Model building and refinement

The DiCas7-11 protein was built from an AlphaFold- (*Jumper et al., 2021*) predicted structure model using the program COOT (*Emsley and Cowtan, 2004*). Both the post-processed and local resolution filtered maps were used in building various parts of the model. The final DiCas7-11-crRNA-target RNA complex was refined in PHENIX (*Liebschner et al., 2019*) to satisfactory stereochemistry and density correlation parameters (*Supplementary file 1*).

## Acknowledgements

This work was supported by NIH grant R01 GM101343 to HL. The authors would like to thank G Seo and S Miller for their support in protein expression and DNA sequencing. The authors also acknowledge the use of instruments at the Biological Science Imaging Resource supported by Florida State University. The Titan was funded from NIH grant S10 RR025080. The BioQuantum/K3 was funded from NIH grant U24 GM116788. The Vitrobot Mk IV was funded from NIH grant S10 RR024564. The Solaris Plasma Cleaner was funded from NIH grant S10 RR024564. The DE-64 was funded from NIH grant U24 GM116788. The Laboratory for BioMolecular Structure (LBMS) is supported by the DOE Office of Biological and Environmental Research (KP160711).

## Additional information

### Funding

| Funder | Grant reference number | Author |
|---|---|---|
| National Institutes of Health | GM101343 | Hong Li |

The funders had no role in study design, data collection and interpretation, or the decision to submit the work for publication.

### Author contributions

Hemant N Goswami, Conceptualization, Validation, Investigation, Writing - original draft, Writing - review and editing; Jay Rai, Validation, Investigation, Writing - review and editing; Anuska Das, Investigation, Writing - review and editing; Hong Li, Conceptualization, Resources, Supervision, Funding acquisition, Investigation, Writing - original draft, Project administration, Writing - review and editing

### Author ORCIDs

Hemant N Goswami (iD) http://orcid.org/0000-0003-4807-9187
Hong Li (iD) http://orcid.org/0000-0003-2046-9861

### Decision letter and Author response

Decision letter https://doi.org/10.7554/eLife.81678.sa1
Author response https://doi.org/10.7554/eLife.81678.sa2

## Additional files

### Supplementary files

- MDAR checklist

- Supplementary file 1. Data acquisition and processing parameters. Related to *Figures 2–4*.

- Supplementary file 2. Ribonucleic acid sequences used in this study. Related to *Figures 1–4*.

### Data availability

Structure model generated from this study is deposited to Protein Data Bank under the accession code 8D1V. The cryoEM map is deposited to EMDB under the accession code EMD-27138.

The following datasets were generated:

| Author(s) | Year | Dataset title | Dataset URL | Database and Identifier |
|---|---|---|---|---|
| Goswami HN, Rai J, Das A, Li H | 2022 | Cryo-EM structure of guide RNA and target RNA bound Cas7-11 | https://www.rcsb.org/structure/8D1V | RCSB Protein Data Bank, 8D1V |
| Goswami HN, Rai J, Das A, Li H | 2022 | Cryo-EM structure of guide RNA and target RNA bound Cas7-11 | https://www.ebi.ac.uk/emdb/EMD-27138 | EMDB, EMD-27138 |

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
