## [Editor Report]

This manuscript is a timely contribution to the CRISPR/Cas field: the mode of function of the type III-E Cas7-11 CRISPR-Cas system. This is an RNA-guided RNA targeting system only characterized last year. In contrast to Cas13 systems, Cas7-11 does not possess collateral damaging activity, hence does not show cytotoxicity when introduced into human cells. These are highly desirable traits in practical applications.

---

## [Decision Letter]

**Decision letter after peer review:**

Thank you for submitting your article "Molecular Mechanism of Active Cas7-11 in Processing CRISPR RNA and Interfering Target RNA" for consideration by *eLife*. Your article has been reviewed by 2 peer reviewers, and the evaluation has been overseen by Timothy Nilsen as Reviewing Editor and James Manley as the Senior Editor. The reviewers have opted to remain anonymous.

*Essential revisions:*

The reviewers thought that the work was interesting and in principle appropriate for *eLife*. Nevertheless, each raised a significant number of concerns regarding the presentation of the work. Please address these concerns as thoroughly as possible. We hope that the concerns can be handled through textual revisions and revision of the figures.

*Reviewer #1 (Recommendations for the authors):*

I have the following points for the authors to address.

1. Comparison with Type III Csm complex – In page 4, the authors claim "the similarity in the bound RNA trajectory at the core between the two complex[es] is striking (Supplementary Figure 4)". However, Figure S4 only showed side-by-side comparison of the complex. There was no superpositioning nor r.m.s.d. value for the readers to evaluate the claims. These need to be provided.

2. "The first 18 nucleotides of the target RNA (+19* ~ +2*) remain base paired with the spacer region of the crRNA (Figure 1c, Supplementary Figure 3, & Supplementary Figure 5)." – What happens to the +1* nucleotide? Is it base paired with the guide? If not, why?

3. Along the same line, the role of the PFS motif in DiCas7-11 should be discussed. While most of the 5´ handle residues are sequestered, the last two nucleotides are displayed by DiCas7-11. The authors should discuss whether their structural observations in this region agree the biochemical data by Ozcan et al. For example, can the authors define the A-1*, G-2*, and the preceding nucleotides in their density map? Is there any interaction between this region and C-1A-2 in the 5´ handle?

4. Along the same line, the authors should include a zoom-in view of each guide RNA-target segment. It is hard to tell from viewing Figure 1C whether the guide and target form perfect, distorted, or shearing base pairing in each segment. Suggesting including a supplementary figure showing the zoomed base pairing densities.

5. I suggest the authors to improve the rendering of the EM maps in supplementary figure 3. For the three enzymatic centers, the important H-bonds should be marked with dashed lines, and distances labeled.

6. Page 7, 5´ handle recognition: as the authors pointed out, not all 15 handle residues are highly conserved among Cas7-11 guide RNAs. Can the authors rationalize the handle nucleotide conservation based on variations in the recognition residues in Cas7-11?

7. Enzymatic mechanism of crRNA biogenesis – Typically besides His, additional polar residues are involved in assisting the proton shuttering. Are there any such residues conserved among Cas7-11s?

8. Cleavage at site 1 – I did not quite follow the author's conclusion that because Y360 is not required for site 1 cleavage, D429 is sufficient for mediating the metal-dependent cleavage. Is Y360 the only alternative catalytic residue in this site? Given that site 1 cleavage is metal independent, D429 is unlikely the general acid/base. Did I misunderstand something here?

9. Figure 3c – No target band, nor cleavage product bands in lanes 4 and 5. Why is that? Are they not labeled? I could not make sense of this panel from reading the text and legend.

10. Figure 3e – Why isn't substrate 5 cleaved but substrate 7 cleaved. Both results seem surprising.

11. Page 5, first sentence of the 2nd paragraph: how does supplementary Figure 5 support the claim of structural similarity?

*Reviewer #2 (Recommendations for the authors):*

1. I found Figure 2 and 3 (especially 3b-f) difficult to follow due to the lack of description of gel visualization methods and the overuse of "asterisk" in the figure legends. The following suggestions may help to reduce such confusion for future readers of this paper:

a. In the biochemical experiments, gels were visualized using either SybrGold staining or fluorescence labeling of the target RNA. The visualization method should be clearly stated in the figure legend for each panel containing a gel, to make it easier for the reader to understand the gel.

b. In some cases, the "target" label on the left side of the gel is not aligned correctly with all targets shown on the gel (e.g. in Figures 3c, 3e), because targets of multiple sizes were used on the same gel. Please adjust these labels.

c. Please describe which substrates were Cy5 labeled in the figure legend for Figure 3b, and/or describe what the yellow circle at the 5' end of target RNA 1 indicates.

d. Please list the target RNAs in Figure 3b in the order they are shown on gels in panels 3c-f.

e. Please describe the blue stars depicting deoxy positions for RNA target 2 in Figure 3b as "blue stars" rather than as asterisks.

f. The use of an asterisk to indicate the labeled target in panel 3d and the spurious RNA in panel 3e is confusing. Could one of these be indicated using a different label? For example, the labeled target RNA could be indicating using the same yellow circle that was used in the cartoon of target RNA 1.

2. Kato et al. described Zinc-finger domains in the Cas7 domains. Is evidence for these domains present in the cryo-EM map obtained in this study? These were not built into the structural model that was provided to reviewers.

3. p. 4 line 15: The authors state that the density map revealed a "partially cleaved target RNA" but do not explain their interpretation of the map that led them to conclude that the RNA is partially cleaved. What was this determination based on? As mentioned in the public review, there appears to be little density for the site 1 product. Further discussion of this interpretation should be included in the manuscript.

4. p. 3 line 22: The authors suggest that Cas7-11 may be an "evolutionary intermediate" between Class 1 and 2 effectors. This terminology is misleading, as it suggests that Class 1 and 2 effectors are homologous. Please reword this sentence.

5. p. 6 line 6: The authors refer to the catalytic mechanism employed for crRNA processing as an "RNase A-like mechanism". A more generic term like "general acid-base mechanism" would be more appropriate, as there are likely several differences between RNase A and the Cas7-11 catalytic activity (e.g. there is only one catalytic histidine in the Cas7-11 active site and two in the RNase A active site; it is unclear whether the cyclic phosphate is resolved via hydrolysis by Cas7-11 as it is by RNase A).

6. p. 7 line 2: The authors compare a structure in the crRNA 5' handle to a 3-10 helix of a protein. It is unclear how these are similar, as this region of the RNA has the base of residue -5 stacked between the bases of residues -7 and -8, which is quite a distinct architecture than what would be observed for the side chains of a 3-10 helix. The phrase about the 3-10 helix could be removed.

7. Please use consistent nomenclature for the Csm complex subunits. It would be helpful to a broad audience to describe Csm3 as the Cas7-like subunit and Csm2 as the Cas11-like subunit in the text (p. 4 lines 20-21). Cas10 is referred to as Csm1 at the top of p. 5. Please use the updated nomenclature of Cas10 only. Csm1 could be indicated in parenthesis when first mentioning Cas10.

8. Reference 1 could be updated to the more recent Makarova review (PMID: 31857715).

9. p. 3 line 4: "efforts" should be "effectors"

10. p. 3 line 11: "Higher Prokaryotic and Eukaryotic binding domain" should be "Higher Eukaryotic and Prokaryotic Nucleotide binding domain"

11. p. 5 line 8: The reference to Supplementary Figure 5 seems extraneous here.

12. p. 6 line 21: Please cite Figure 2a at the end of this sentence.

13. p. 8 line 18: The authors state that Asp429 is sufficient for mediating metal-dependent cleavage, but the paragraph presents data that target cleavage in this active site is metal independent. This sentence is incongruous with the rest of the paragraph and requires clarification.

14. A general comment regarding the structure figures: The figures are very attractive, but it would be helpful if the side chains of proteins were colored by heteroatom to more readily observe the identity of the residue and potential polar contacts.

15. In Supplementary Figure 1, please indicate which peak was collected for the bottom elution profile.

16. In Supplementary Figure 2, does "pcts" stand for "particles"? It may be helpful to spell out the whole word, as this is not a standard abbreviation as far as I know.

17. In Supplementary Figure 4, it would be helpful to show Cas7-11 in this figure so that the reader does not need to compare with a separate figure.

---

## [Author Response]

Reviewer #1 (Recommendations for the authors):I have the following points for the authors to address.1. Comparison with Type III Csm complex – In page 4, the authors claim "the similarity in the bound RNA trajectory at the core between the two complex[es] is striking (Supplementary Figure 4)". However, Figure S4 only showed side-by-side comparison of the complex. There was no superpositioning nor r.m.s.d. value for the readers to evaluate the claims. These need to be provided.

We revised Figure S4 to include three views of superimposed structure images as the new panel b and the associated r.m.s.d in the figure legend.

2. "The first 18 nucleotides of the target RNA (+19* ~ +2*) remain base paired with the spacer region of the crRNA (Figure 1c, Supplementary Figure 3, & Supplementary Figure 5)." – What happens to the +1* nucleotide? Is it base paired with the guide? If not, why?

We should have clearly labeled the nucleotides that are modeled based on density, which does not include +1*. We now include in Figure 1 caption to indicate that the gray colored nucleotides, though included in the RNA constructs, are not modeled due to weak or no density.

3. Along the same line, the role of the PFS motif in DiCas7-11 should be discussed. While most of the 5´ handle residues are sequestered, the last two nucleotides are displayed by DiCas7-11. The authors should discuss whether their structural observations in this region agree the biochemical data by Ozcan et al. For example, can the authors define the A-1*, G-2*, and the preceding nucleotides in their density map? Is there any interaction between this region and C-1A-2 in the 5´ handle?

Unfortunately, our model does not include any of the PFS nucleotides. However, we did not do a good job in indicating which nucleotides are modeled (see above). We now include in Figure 1 caption to indicate that the gray colored nucleotides, though included in the RNA constructs, are not modeled due to weak or no density.

Though we did not observe any of the PFS nucleotides mentioned by the reviewers, we do have an opportunity to speculate the roles of PFS given the observed 5’-tag structure. We added a statement on page 8 speculating that the two unpaired (C-1 and A-2) could pair with PFS to influence Cas7-11 structure and its downstream activity such as activation of TPR-CHAT by Hu et al.

“The different mode of 5’ tag binding to Cas7-11 from that of Csm explains the lack of a Cas10-like effector as in Csm. Rather, it allosterically activates a different effector, the TPRCHAT protease, in a 3’ PFS-dependent manner. Though 13 of the 15 5’-tag nucleotides are buried by the protein, the last two, C(-1) and A(-2), are partially solvent-accessible and can potentially participate in PFS-mediated activation of the TRP-CHAT activity ^20^.”

4. Along the same line, the authors should include a zoom-in view of each guide RNA-target segment. It is hard to tell from viewing Figure 1C whether the guide and target form perfect, distorted, or shearing base pairing in each segment. Suggesting including a supplementary figure showing the zoomed base pairing densities.

We added additional close-up views of density of the guide RNA-target pairings in Figure S3. In addition, we added an image of superimposed guide RNA-target helix on a standard A-form helix.

5. I suggest the authors to improve the rendering of the EM maps in supplementary figure 3. For the three enzymatic centers, the important H-bonds should be marked with dashed lines, and distances labeled.

Thanks for the suggestion. The EM maps and the close contacts are now updated for Figure S3.

6. Page 7, 5´ handle recognition: as the authors pointed out, not all 15 handle residues are highly conserved among Cas7-11 guide RNAs. Can the authors rationalize the handle nucleotide conservation based on variations in the recognition residues in Cas7-11?

We revised the paragraph in page 7 extensively and with careful citations to supplementary figure 6 (now Figure 2—figure supplementary 1) to reflect the fact that “In general, well-conserved residues form base-specific interactions while non-conserved residues maintain interactions with RNA sugar phosphate backbone.”. We clearly state if an amino acid is conserved or not when describing its mode of interactions. We explicitly point out that the non-conserved RNA nucleotides, A(-3), C(-6), and C(-8), form minimal contacts with proteins.

7. Enzymatic mechanism of crRNA biogenesis – Typically besides His, additional polar residues are involved in assisting the proton shuttering. Are there any such residues conserved among Cas7-11s?

We appreciate this suggestion. We now extend our discussion to include discussion of the surrounding residues that can potentially influence processing activity.

8. Cleavage at site 1 – I did not quite follow the author's conclusion that because Y360 is not required for site 1 cleavage, D429 is sufficient for mediating the metal-dependent cleavage. Is Y360 the only alternative catalytic residue in this site? Given that site 1 cleavage is metal independent, D429 is unlikely the general acid/base. Did I misunderstand something here?

In the original manuscript, we meant to use the phrase “metal-independent” instead of “metal-dependent” because at the time, we were convinced this site uses a metalindependent mechanism, which prompted us to look for possible general acid/base residues such as the well-conserved Tyr360.

This conclusion had since been revised given the new data described below (see response to item 3 of Reviewer 2). In an effort of repeating the target cleavage assay, we discovered that two different batches of DiCas7-11 resulted in different metal-dependence. Whereas the older batch used for biochemistry studies such as those in Figure 3 reproducibly produces metalindependent cleavage of site 1, the batch later purified showed a complete dependence on metal (revised Figure 3). The two batches were from two different clones, one was constructed in-house and another described by Ozcan et al. and purchased from Addgene (Özcan, Ahsen, et al. “Programmable RNA targeting with the single-protein CRISPR effector Cas711.” *Nature* 597.7878 (2021): 720-725.). Though both encode an identical protein with affinity tag differences, the in-house clone produced samples with noticeably more degraded or prematurely terminated proteins (see figure in and response to item 3 of Reviewer 2). Though the cleavage gel patterns from the in-house clone-derived protein resemble those by Ozcan et al. that seem to suggest metal independence for site 1, the cleaner protein prep from the Addgene clone clearly showed metal dependent cleavage for both sites (see figure in and response to item 3 of Reviewer 2). We thus updated our target cleavage results and discussions throughout the manuscript to reflect this new result. The exact reason for the different cleavage patterns with different proteins is not clear at this time (see figure in and response to item 3 of Reviewer 2).

Therefore, the nonessential role of Y360 for site 1 can now be better explained, given the fact that Asp429 is likely sufficient for metal-dependent cleavage. The previous statement is in fact appropriate now.

9. Figure 3c – No target band, nor cleavage product bands in lanes 4 and 5. Why is that? Are they not labeled? I could not make sense of this panel from reading the text and legend.

Thank for pointing this out. The reason is that crRNA is coincidentally the same size as the deoxy-modified target RNA used in this assay. To avoid the confusion, we reanalyzed the same reactions by using a shorter deoxy target RNA and updated Figure 3c. The conclusion is now clear.

10. Figure 3e – Why isn't substrate 5 cleaved but substrate 7 cleaved. Both results seem surprising.

The result is correct. However, we should have made more clear what exactly these target RNA are. In the revised Figure 3, we clearly indicate the length and possible cleavage sites of each target RNA. The original substrate 5 (now 7), though flanking both cleavage sites, has only 12-base pairs with crRNA, which could be insufficient for substrate binding and is thus not cleaved. In contrast, substrate 7 (now 5), though flanks only site 2, can base pair with 18 crRNA nucleotides and is thus cleaved.

Given the fact that substrate 6 is also cleaved (16-base pairs), we suggest that target RNA pairing with either end of the crRNA for at least 16 bases are sufficiently cleaved.

11. Page 5, first sentence of the 2nd paragraph: how does supplementary Figure 5 support the claim of structural similarity?

We mistakenly cited Supplementary Figure 5 here. It is now deleted.

Reviewer #2 (Recommendations for the authors):1. I found Figure 2 and 3 (especially 3b-f) difficult to follow due to the lack of description of gel visualization methods and the overuse of "asterisk" in the figure legends. The following suggestions may help to reduce such confusion for future readers of this paper:a. In the biochemical experiments, gels were visualized using either SybrGold staining or fluorescence labeling of the target RNA. The visualization method should be clearly stated in the figure legend for each panel containing a gel, to make it easier for the reader to understand the gel.

Thank you for the suggestion. We have revised all figures to contain consistent symbols indicating either processing or target cleavage sites, various types of RNA used and the cleavage products. In addition, we also found a way to distinguish the two different types of staining used for various target RNA (see updated Figure 3).

b. In some cases, the "target" label on the left side of the gel is not aligned correctly with all targets shown on the gel (e.g. in Figures 3c, 3e), because targets of multiple sizes were used on the same gel. Please adjust these labels.

See item a.

c. Please describe which substrates were Cy5 labeled in the figure legend for Figure 3b, and/or describe what the yellow circle at the 5' end of target RNA 1 indicates.

See item a.

d. Please list the target RNAs in Figure 3b in the order they are shown on gels in panels 3c-f.

See item a.

e. Please describe the blue stars depicting deoxy positions for RNA target 2 in Figure 3b as "blue stars" rather than as asterisks.

See item a.

f. The use of an asterisk to indicate the labeled target in panel 3d and the spurious RNA in panel 3e is confusing. Could one of these be indicated using a different label? For example, the labeled target RNA could be indicating using the same yellow circle that was used in the cartoon of target RNA 1.

See item a.

2. Kato et al. described Zinc-finger domains in the Cas7 domains. Is evidence for these domains present in the cryo-EM map obtained in this study? These were not built into the structural model that was provided to reviewers.

Thanks for pointing this out. We did neglect the zinc atoms at these four Zinc fingers that are now added to the revised coordinate and the revised Figure 1.

3. p. 4 line 15: The authors state that the density map revealed a "partially cleaved target RNA" but do not explain their interpretation of the map that led them to conclude that the RNA is partially cleaved. What was this determination based on? As mentioned in the public review, there appears to be little density for the site 1 product. Further discussion of this interpretation should be included in the manuscript.

The “partially cleaved target RNA” phrase is a result of weak density for nucleotides beyond site 1 (+2* and +3*) but clear density flanking site 2, despite the presence of the active enzyme. This feature indicates that cleavage likely had taken place at site 1 but not site 2 in most of the particles went into the reconstruction. To further support this phrase, we added “The PFS region plus the first base paired nucleotide (+1*) are not observed.”

4. p. 3 line 22: The authors suggest that Cas7-11 may be an "evolutionary intermediate" between Class 1 and 2 effectors. This terminology is misleading, as it suggests that Class 1 and 2 effectors are homologous. Please reword this sentence.

We revised the phrase to “Cas7-11 is therefore believed to be a unique Type III CRIPSR-Cas system.” to avoid the confusion.

5. p. 6 line 6: The authors refer to the catalytic mechanism employed for crRNA processing as an "RNase A-like mechanism". A more generic term like "general acid-base mechanism" would be more appropriate, as there are likely several differences between RNase A and the Cas7-11 catalytic activity (e.g. there is only one catalytic histidine in the Cas7-11 active site and two in the RNase A active site; it is unclear whether the cyclic phosphate is resolved via hydrolysis by Cas7-11 as it is by RNase A).

This statement is indeed speculative, and we agree there are likely differences between RNase A and Cas7-11. Without thorough enzyme kinetic studies, the catalytic mechanism of Cas7-11 remains at large. We do not know if His43 acts as general acid or base, nor do we know if there are other amino acids nearby that participate in rate enhancement. We revised the statement to “The requirement for His43 without requiring divalent metals in processing suggests that the C7L.1 employs a general acid-base catalysis mechanism similarly suggested for Cas6 ^16,17^, although detailed roles of His43 and other possible catalytic residues remain to be characterized.”

6. p. 7 line 2: The authors compare a structure in the crRNA 5' handle to a 3-10 helix of a protein. It is unclear how these are similar, as this region of the RNA has the base of residue -5 stacked between the bases of residues -7 and -8, which is quite a distinct architecture than what would be observed for the side chains of a 3-10 helix. The phrase about the 3-10 helix could be removed.

The reviewer is correct that we need further clarify the reference to 3-10 helix. We should have emphasized that the comparison is between the backbone of the two kinds of polymers, especially the intra-chain interactions established among the backbone atoms.

We revised the statement to “The tight and the N+3 interactions involving the C(-8)-A(-7)C(-6)-G(-5) backbone atoms is reminiscent of those within a 3_10_ helix in proteins.”. This statement is also better positioned in the text after a series of descriptions of the unusual tight turn interactions.

7. Please use consistent nomenclature for the Csm complex subunits. It would be helpful to a broad audience to describe Csm3 as the Cas7-like subunit and Csm2 as the Cas11-like subunit in the text (p. 4 lines 20-21). Cas10 is referred to as Csm1 at the top of p. 5. Please use the updated nomenclature of Cas10 only. Csm1 could be indicated in parenthesis when first mentioning Cas10.

Thanks for the suggestion. We revised the statement to “The entire DiCas7-11 complex can be superimposed onto the closely matched homologous *Lactococcus lactis* Csm (LlCsm) complex ^13^ with the C7L.1-C7L.4 ridge matching that formed by four Cas7-like (Csm3) subunits and C11L matching the Cas11-like (Csm2) subunit adjacent to Cas10 (Csm1) (Figure 1—figure supplementary 4).”

8. Reference 1 could be updated to the more recent Makarova review (PMID: 31857715).

We have updated reference 1.

9. p. 3 line 4: "efforts" should be "effectors"

Corrected.

10. p. 3 line 11: "Higher Prokaryotic and Eukaryotic binding domain" should be "Higher Eukaryotic and Prokaryotic Nucleotide binding domain"

Corrected.

11. p. 5 line 8: The reference to Supplementary Figure 5 seems extraneous here.

Corrected.

12. p. 6 line 21: Please cite Figure 2a at the end of this sentence.

Corrected.

13. p. 8 line 18: The authors state that Asp429 is sufficient for mediating metal-dependent cleavage, but the paragraph presents data that target cleavage in this active site is metal independent. This sentence is incongruous with the rest of the paragraph and requires clarification.

Corrected, and please also see the response to Reviewer 1’s #8 comment.

14. A general comment regarding the structure figures: The figures are very attractive, but it would be helpful if the side chains of proteins were colored by heteroatom to more readily observe the identity of the residue and potential polar contacts.

We have replaced the figures with heteroatom colored side chains of proteins.

15. In Supplementary Figure 1, please indicate which peak was collected for the bottom elution profile.

We labeled the peak now.

16. In Supplementary Figure 2, does "pcts" stand for "particles"? It may be helpful to spell out the whole word, as this is not a standard abbreviation as far as I know.

We replaced “pcts” by “particles”.

17. In Supplementary Figure 4, it would be helpful to show Cas7-11 in this figure so that the reader does not need to compare with a separate figure.

A new panel (b) showing superimposed Cas7-11 and LlCsm is now added to Figure S4.